# Conditional Diffusion Models are Medical Image Classifiers that Provide Explainability and Uncertainty for Free

**Gian Mario Favero**$^*$                                            GIAN.FAVERO@MAIL.MCGILL.CA
**Parham Saremi**$^*$                                                PARHAM.SAREMI@MAIL.MCGILL.CA
**Emily Kaczmarek**                                              EMILY.KACZMAREK@MAIL.MCGILL.CA
**Brennan Nichyporuk**                                               NICHYPOB@MILA.QUEBEC
**Tal Arbel**                                                         TAL.ARBEL@MCGILL.CA
*Center for Intelligent Machines, McGill University, Montreal, Canada.*
*Mila - Quebec AI Institute, Montreal, Canada.*

**Editors:** Accepted for publication at MIDL 2025

## Abstract

Discriminative classifiers have become a foundational tool in deep learning for medical imaging, excelling at learning separable features of complex data distributions. However, these models often need careful design, augmentation, and training techniques to ensure safe and reliable deployment. Recently, diffusion models have become synonymous with generative modeling in 2D. These models showcase robustness across a range of tasks including natural image classification, where classification is performed by comparing reconstruction errors across images generated for each possible conditioning input. This work presents the first exploration of the potential of class conditional diffusion models for 2D medical image classification. First, we develop a novel majority voting scheme shown to improve the performance of medical diffusion classifiers. Next, extensive experiments on the CheXpert and ISIC Melanoma skin cancer datasets demonstrate that foundation and trained-from-scratch diffusion models achieve competitive performance against SOTA discriminative classifiers without the need for explicit supervision. In addition, we show that diffusion classifiers are intrinsically explainable, and can be used to quantify the uncertainty of their predictions, increasing their trustworthiness and reliability in safety-critical, clinical contexts. Further information is available on our project page: https://faverogian.github.io/med-diffusion-classifier.github.io/.

**Keywords:** diffusion, classification, explainability, uncertainty

## 1. Introduction

Deep learning applications in medicine have received significant attention in recent years due to their potential to revolutionize healthcare outcomes. For instance, the ability to accurately classify disease pathology from medical images using discriminative classifiers (e.g., ResNet (He et al., 2015), ViT (Dosovitskiy et al., 2020)) is central to advancing early diagnosis, personalized treatment, and overall patient care. In the ideal scenario, discriminative classifiers are robust and generalizable; however, state-of-the-art performance often relies heavily on data augmentation and hyperparameter tuning, which can be time- and computation-expensive, and may still be prone to overfitting and/or learning short-cuts (Geirhos et al., 2020). Even with strong classification performance, models must be

---

$^*$ Contributed equally

explainable and provide uncertainty estimates to ensure reliable and trustworthy predictions for safe clinical deployment. Current explainability and uncertainty methods depend largely on post-hoc analysis or model modifications. For example, explainability often relies on gradient-based analysis after training (Selvaraju et al., 2020) or counterfactual generation with a separate model (Sun et al., 2023), whereas uncertainty methods range from simple model modifications, like Monte Carlo (MC) dropout, to expensive ensembling methods. Thus, there remains limitations to the safe use of discriminative classifiers in medical imaging, particularly due to the lack of built-in explainability and uncertainty analysis.

Diffusion models (Ho et al., 2020) make up one class of generative models that has shown remarkable flexibility and robustness across various deep learning tasks, achieving state-of-the-art performance in image (Dhariwal and Nichol, 2021), video (Ho et al., 2022), and audio (Kong et al., 2020) generation tasks. Recently, generative models have been used directly for image classification (Li et al., 2023; Clark and Jaini, 2023; Krojer et al., 2023; Chen et al., 2024) in natural imaging, showing that large pre-trained models like Stable Diffusion (Rombach et al., 2022) can be used as classifiers that are competitive with state-of-the-art supervised discriminative classifiers (He et al., 2015; Dosovitskiy et al., 2020). Diffusion models are increasingly being used in the medical domain for data augmentation (Guo et al., 2024), segmentation (Wu et al., 2023a,b; Amit et al., 2023), anomaly detection (Wolleb et al., 2022), counterfactual explanation (Sanchez et al., 2022; Bedel and Çukur, 2023; Weng et al., 2024; Pegios et al., 2024), and probabilistic classification (Shen et al., 2024). However, despite many conditional diffusion models developed for medical image analysis, they have yet to be explored as classifiers that can provide explainability and uncertainty-estimation for free.

In this work, we present a comprehensive evaluation of how conditional diffusion models can be re-purposed and leveraged for image classification, explainability, and uncertainty estimation in the medical domain. First, we propose a novel majority voting-based method that improves the performance of diffusion classifiers in medical imaging. We then demonstrate that classifiers derived from foundation and trained-from-scratch diffusion models perform competitively with state-of-the-art medical image discriminative classifiers through extensive experiments on the publicly available CheXpert (Irvin et al., 2019) and ISIC Melanoma skin cancer (Rotemberg et al., 2020) datasets, despite not being trained for classification. Next, we show that diffusion classifiers offer explainability (via counterfactual generation) and uncertainty quantification (via entropy) out-of-the-box. We validate the uncertainty by showing that when the model is confident, it is correct, and vice versa. This is shown as model accuracy drastically improves as its uncertainty threshold increases. For example, Stable Diffusion reaches classification accuracies of 100% and 95% on ISIC and CheXpert, respectively, with only 45% of its most uncertain samples filtered out.

## 2. Methodology

In this work, we present diffusion classifiers for medical imaging classification tasks. We first present an overview of diffusion models in Section 2.1. Next, in Section 2.2, we define conditional diffusion models and demonstrate how they can perform classification. Section 2.3 introduces all extensions to the diffusion classifier, including: our novel algorithm for im-

proving classification performance through majority voting, as well as the ability to perform counterfactual explainability and uncertainty quantification without any modifications.

## 2.1. Diffusion Models

Diffusion models (DM) are likelihood-based models that learn to approximate a data distribution through a process of iterative noising and denoising involving two key phases: a fixed forward process and a learned backward process. In the forward process, Gaussian noise $\epsilon \sim \mathcal{N}(0, \mathrm{I})$ is gradually added to data in a controlled manner, destroying its structure until it is pure Gaussian noise.

This process, which is done on a sample over time, can be expressed by its marginal for all $t$ on a continuous interval, $[0, 1]$:

$$q(\boldsymbol{z}_t|\boldsymbol{x}) = \alpha_\lambda \boldsymbol{x} + \sigma_\lambda \boldsymbol{\epsilon} \text{ where } \boldsymbol{\epsilon} \sim \mathcal{N}(0, \mathrm{I}). \tag{1}$$

Following the variational diffusion model formulation (Kingma et al., 2023), the forward process is defined to be variance-preserving, imposing the constraint $\alpha_\lambda^2 = \mathrm{sigmoid}(\lambda)$, $\sigma_\lambda^2 = \mathrm{sigmoid}(-\lambda)$, where $\lambda$ is the log-SNR given by $\lambda = f_\lambda(t)$ and $f_\lambda(t)$ is the noise schedule (see Appendix A.1). The noise schedule is a monotonically decreasing and invertible function that connects the time variable, $t$, with the log-SNR, $\lambda$. During training, $t$ is sampled from a continuous, uniform distribution, $\mathcal{U}(0, 1)$, which is then used to compute $\lambda$. The resulting distribution over noise levels can be defined as $p(\lambda) = -1/f'_\lambda(t)$ (Kingma and Gao, 2023).

In the backward process, a neural network attempts to learn how to remove the added noise and recover an approximate sample from the original data distribution. Kingma et al. show that the variational lower bound objective (VLB) function for training diffusion models can be derived in continuous time with respect to its log-SNR, $\lambda$, noise sampling distribution, $p(\lambda)$ and weighting function, $w(\lambda)$ (Kingma et al., 2023). This VLB is:

$$\log p(x) = \mathcal{L}_x + \mathcal{L}_T - \mathbb{E}_{\epsilon \sim \mathcal{N}(0, \mathrm{I}), \lambda \sim p(\lambda)} \left[ \frac{w(\lambda)}{p(\lambda)} ||\boldsymbol{x} - \hat{\boldsymbol{x}}_\theta(\boldsymbol{z}_\lambda; \lambda)||_2^2 \right]. \tag{2}$$

Where $\mathcal{L}_x = -\log p(\boldsymbol{x}|\boldsymbol{z}_0) \approx 0$ for discrete $\boldsymbol{x}$ and $\mathcal{L}_T = D_{KL}(q(\boldsymbol{z}_T|x)||p(\boldsymbol{z}_T)) \approx 0$ for a well-defined forward process. We use a min-SNR weighting function (Hang et al., 2024), a shifted-cosine noise schedule (Hoogeboom et al., 2023), and v-prediction parameterization for greater stability during training and sampling (Salimans and Ho, 2022).

## 2.2. Conditional Diffusion Models as Classifiers

Conditional diffusion models incorporate text or categorical inputs, such that the prediction becomes $\hat{\boldsymbol{x}}_\theta(\boldsymbol{z}_\lambda, c)$ where $c$ is a conditioning embedding. In this paper, we implement conditioning through cross-attention in a UNet-based diffusion model (Rombach et al., 2022), and adaptive layer normalization in DiTs (Peebles and Xie, 2023).

Recent works (Li et al., 2023; Clark and Jaini, 2023; Krojer et al., 2023; Chen et al., 2024) have explored using conditional diffusion models as discriminative classifiers. As shown in Figure 1, classification is performed by comparing reconstruction errors across images generated for each possible conditioning input. Specifically, using the labels, $\boldsymbol{C} = \{c_i\}$, and

Bayes' theorem on model predictions, $p(\boldsymbol{x}|c_i)$, we can derive $p(c_i|\boldsymbol{x})$:

$$p(c_i|\boldsymbol{x}) \approx \frac{\exp\{\mathbb{E}_{\epsilon\sim\mathcal{N}(0,\mathrm{I}),\lambda\sim p(\lambda)}\left[||\boldsymbol{x} - \hat{\boldsymbol{x}}_\theta(\boldsymbol{z}_\lambda, c_i)||_2^2\right]\}}{\exp\{\sum_j \mathbb{E}_{\epsilon\sim\mathcal{N}(0,\mathrm{I}),\lambda\sim p(\lambda)}\left[||\boldsymbol{x} - \hat{\boldsymbol{x}}_\theta(\boldsymbol{z}_\lambda, c_j)||_2^2\right]\}}. \tag{3}$$

A more complete derivation is found in Appendix A.2. A Monte Carlo estimation of the expectation for an arbitrary class, $c_j$, can be computed by sampling $N$ noise level pairs, $(\boldsymbol{\epsilon}, \lambda)$ and averaging the reconstruction error:

$$\frac{1}{N}\sum_{k=1}^{N}\left[||\boldsymbol{x} - \hat{\boldsymbol{x}}_\theta(\alpha_{\lambda_k}\boldsymbol{x} + \sigma_{\lambda_k}\boldsymbol{\epsilon}_k, c_j)||_2^2\right]. \tag{4}$$

For each $(\boldsymbol{\epsilon}, \lambda)$ pair, $\boldsymbol{\epsilon}$ is sampled from an isotropic Gaussian distribution and $\lambda$ is sampled from $p(\lambda)$ (practically speaking, $t \sim \mathcal{U}(0,1)$, then $\lambda = f_\lambda(t)$). Eq. (4) shows that classifying one sample requires $N$ many steps per condition, where the Monte Carlo estimate becomes more accurate as the number of steps increases. To reduce the variance of prediction for a given image, $\boldsymbol{x}$, an identical set of $(\boldsymbol{\epsilon}_k, \lambda_k) \in S\{(\boldsymbol{\epsilon}_k, \lambda_k)\}_{k=1}^N$ is used for every condition, which increases the accuracy of the prediction $p(\boldsymbol{C}|\boldsymbol{x})$. In practice, Eq. (3) is equivalent to choosing the class with the minimum average reconstruction error.

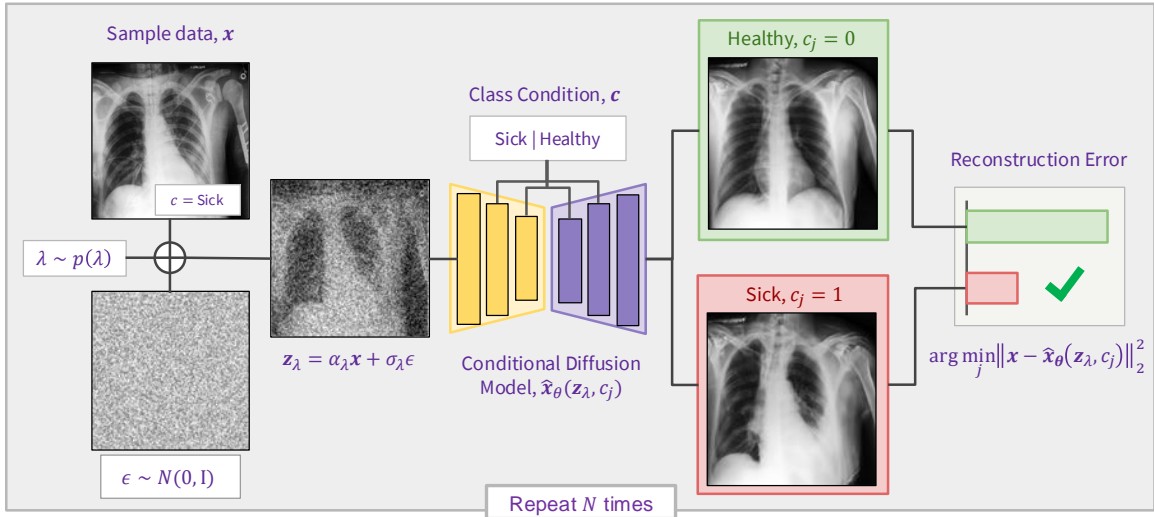

Figure 1: **Diffusion classifiers** are conditional diffusion models repurposed as classifiers. First, a sample, $\boldsymbol{x}$, is noised at a randomly chosen noise level, $(\boldsymbol{\epsilon}_k, \lambda_k)$. The noised sample is then denoised by the diffusion network with each possible conditioning input, $c_j$. The conditioning variable, $c_j$, that results in the denoised output, $\hat{\boldsymbol{x}}_\theta(\boldsymbol{z}_\lambda, c_j)$, with the smallest reconstruction error over many noise levels is selected as the class. This process is repeated for a set of $N$ noise levels $(\boldsymbol{\epsilon}, \lambda)$ with the reconstruction errors aggregated (e.g., average/majority voting) for a more accurate prediction.

### 2.3. Extensions on Diffusion Classifiers

**Majority Voting:** In this work, we introduce a novel majority-vote-based algorithm for determining the predicted class. Here, we tally votes across all $(\epsilon, \lambda)$ pairs by identifying the class with the lowest error as the prediction for that pairing, and take the final predicted class as the one with the majority of individual votes (see Appendix A.3 for the algorithm's pseudocode). We posit that averaging reconstruction error over all noise levels inherently weights higher values of $\lambda$ more, which is not always beneficial (i.e., reconstructions at higher values of $\lambda$ are naturally much harder and thus have greater error, introducing more noise into the average reconstruction error).

**Intrinsic Explainability:** Diffusion classifiers use Classifier-Free Guidance (CFG) (Ho and Salimans, 2022) to understand which features are most influential in generating certain classes. CFG is a common approach in which a conditional diffusion model is simultaneously trained for an unconditional task by randomly dropping out $c$ ($\sim$10% of the time). In doing so, sampling can be guided towards an intended class with a guidance-scale, $w$:

$$\tilde{\boldsymbol{x}}_\theta(\boldsymbol{z}_\lambda, c) = (1 + w)\hat{\boldsymbol{x}}_\theta(\boldsymbol{z}_\lambda, c) - w\hat{\boldsymbol{x}}_\theta(\boldsymbol{z}_\lambda, \varnothing). \tag{5}$$

At inference, the model permits explainability for free, through the conditional generation of the factual and counterfactual images of the input image. First, noise is added to obscure the input images, while preserving enough image structure to make reconstruction possible. Then, by varying the condition at inference, the model can shift its generation process to any possible conditional class. These generated images represent the reconstruction of the image provided by the true class, and any counterfactual image(s), where difference maps can be created to highlight class-specific regions modified by the network.

**Uncertainty Quantification:** Diffusion classifiers are also able to produce uncertainty estimates without any additional modifications to the model. The set of $N$ $(\epsilon, \lambda)$ pairs required for accurate classification results (Eq. (4)) results in numerous predictions generated for each sample and thus inherently resembles the uncertainty estimation strategy via MC dropout or ensemble methods. As explained in Section 2.2, to achieve accurate classification, a single sample requires $N$ steps (repeated per condition) where reconstruction errors for that sample are calculated at different $(\epsilon, \lambda)$ noise levels (Eq. (4)). To quantify the uncertainty of the overall predicted class, we construct a Bernoulli distribution from each of the $N$ predictions. This creates a probability density from which uncertainty can be computed.

## 3. Experiments

We first evaluate the performance of the average reconstruction error objective (Section 2.2) against a majority voting alternative, demonstrating that the latter yields superior results in our tasks. We then compare the classification performance of diffusion classifiers with state-of-the-art discriminative baselines, and furthermore, show that conditional diffusion models are interpretable out-of-the-box, and capable of producing uncertainty quantifications.

| Method | CheXpert | | ISIC | |
|---|---|---|---|---|
| | Majority | Average | Majority | Average |
| DiT-B/4 | $86.8 \pm 0.38$ | $85.3 \pm 0.40$ | $90.5 \pm 0.08$ | $90.4 \pm 0.29$ |
| UNet | $84.6 \pm 0.21$ | $83.4 \pm 0.14$ | $91.2 \pm 0.08$ | $84.3 \pm 0.75$ |
| Stable Diffusion | $84.7 \pm 0.34$ | $79.9 \pm 0.29$ | $94.7 \pm 0.14$ | $94.5 \pm 0.33$ |

Table 1: **Majority voting outperforms averaging** in achieving highest classification accuracy on the CheXpert and ISIC Melanoma test sets (using 501 steps). Model variance is calculated at inference based on five random seeds.

### 3.1. Datasets

**ISIC Melanoma:** A publicly available dataset (Rotemberg et al., 2020) containing over 35,000 images of skin lesions and corresponding labels for the presence of melanoma. We balance the dataset by class for our experiments, resulting in 10,212 total images. The data are randomly split into an 80/10/10 train/validation/test set.

**CheXpert:** A publicly available dataset (Irvin et al., 2019) containing over 200,000 chest X-ray images with binary labels for 14 diseases and the presence of support devices. For our experiments, we use "Pleural Effusion" and "No Findings" as mutually exclusive labels and filter for frontal views of the chest, resulting in a balanced dataset of 20,404 samples. The data are randomly split into an 80/10/10 train/validation/test set.

### 3.2. Model Architectures

**Baselines:** To establish a comparative baseline, we evaluate the performance of both convolutional and transformer-based architectures. We use `torchvision` implementations of ResNet-18 and ResNet-50 (He et al., 2015), and `timm` implementations of ViT-S/16 and ViT-B/16 (Dosovitskiy et al., 2020), EfficientNet-B0 and EfficientNet-B4 (Tan and Le, 2019), and Swin-B Transformer (Liu et al., 2021).

**Conditional Diffusion Models:** We implement a UNet backbone based on the ADM architecture (Dhariwal and Nichol, 2021) at $256^2$ resolution, incorporating improvements from simple diffusion (Hoogeboom et al., 2023), such as scaling the number of ResBlocks at lower resolutions to save memory at higher resolutions. For transformer-based diffusion models, we include the DiT-B/4 variant from (Peebles and Xie, 2023). Unless otherwise noted, all images are compressed with a single-stage discrete wavelet transform (DWT) using a Haar wavelet to improve computational efficiency.

**Foundation Models:** Ideally, foundation models like Stable Diffusion can be repurposed as zero-shot classifiers. However, we find that such models are not trained on enough medical data to perform adequately by default. Thus, to ensure a fair comparison, we fine-tune Stable Diffusion v2-base (Rombach et al., 2022) on an amalgamation of our CheXpert and ISIC Melanoma training splits. Given that the model is designed for text-to-image generation, we replace labels in the datasets with text prompts, e.g., "a benign skin lesion", or "a

frontal chest xray of a sick patient with pleural effusion". More details on all architectures can be found in Appendices B and C.

## 4. Results

### 4.1. Ablating on the Classification Algorithm

We propose a simple but effective majority voting scheme that, instead of accumulating errors at each timestep, tallies the amount of times a reconstruction error was smaller for each test condition and then chooses the class with the most votes. Table 1 shows that the highest classification accuracy is consistently achieved with majority voting. This result is intuitive: at greater values of $N$ there are more reconstructions attempted from high noise disturbance which can introduce large sources of variance in the average error. Figure 2($a$) shows an ablation study of accuracy against classification steps on the CheXpert and ISIC validation sets when a majority vote algorithm is used. In general, more classification steps lead to better performance, though with diminishing returns. Thus, we use majority voting with 501 steps for all diffusion-based classification results in this paper.

|  | Method | CheXpert | | ISIC | |
| --- | --- | --- | --- | --- | --- |
|  |  | Accuracy | F1 | Accuracy | F1 |
| CNN | ResNet-18 | 90.9 | 0.910 | 94.4 | 0.943 |
|  | ResNet-50 | 91.6 | 0.914 | 93.6 | 0.935 |
|  | EfficientNet-B0 | 90.5 | 0.907 | 93.1 | 0.930 |
|  | EfficientNet-B4 | 90.4 | 0.904 | 93.2 | 0.930 |
| TF | ViT-S/16 | 86.9 | 0.869 | 95.0 | 0.949 |
|  | ViT-B/16 | 85.1 | 0.857 | 94.8 | 0.948 |
|  | Swin-B | 86.1 | 0.863 | 95.9 | 0.958 |
| DM | DiT-B/4 | 86.1 | 0.860 | 90.4 | 0.901 |
|  | UNet | 84.5 | 0.854 | 91.8 | 0.919 |
|  | Stable Diffusion* | 85.0 | 0.839 | 94.8 | 0.946 |
|  | Stable Diffusion† | 48.8 | 0.656 | 39.7 | 0.521 |

Table 2: **Diffusion classifiers are competitive with discriminative baselines.** * and † denote fine-tuned and zero-shot versions, respectively. Results are reported on the CheXpert and ISIC Melanoma test sets, with 501 classification steps and majority voting being used for the diffusion classifiers (DM).

### 4.2. Classification Performance on Benchmark Datasets

Table 2 shows the classification accuracy and F1-score of each model on the CheXpert and ISIC Melanoma test sets. Note that the models are grouped by architecture: convolution-based (CNN), transformer-based (TF), and diffusion-based (DM). These results demonstrate that the diffusion classifier achieves competitive performance with discriminative

baselines. However, unlike other classifiers, the diffusion classifier requires minimal hyper-parameter tuning, no data augmentations, and only a simple and stable MSE loss function during training. A comparison of optimization settings is found in Appendix B.

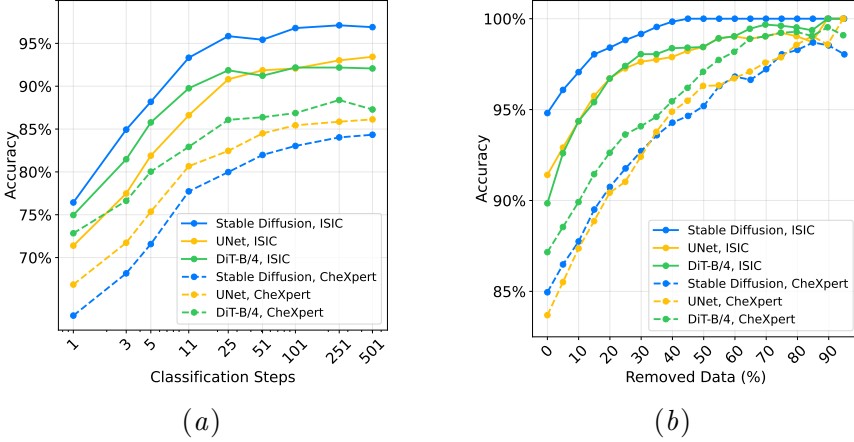

$(a)$ $\qquad$ $(b)$

Figure 2: (a) Ablation on corresponding validation sets demonstrates that higher perfor-
mance comes with more classification steps. (b) Diffusion classifiers inherently
produce uncertainty estimates. Filtering uncertain predictions improves perfor-
mance on the remaining data.

### 4.3. Intrinsic Explainability

A key advantage of diffusion classifiers lies in their intrinsic interpretability, which positions diffusion classifiers as not only effective but also transparent. Importantly, diffusion classifiers are able to produce counterfactual explanations, as opposed to other interpretability methods that simply highlight regions of interest. This can be seen in Figure 3: On the left example (skin lesion), the counterfactual of a malignant lesion (melanoma) has changed colour and intensity to become healthy. In the right example (chest X-ray), the counterfactual image of a sick patient (pleural effusion) shows decreased disease pathology in the left and right lungs. The natural interpretability of diffusion classifiers provides both trans-parency on how the model is learning (thus allowing the identification of shortcut learning), and specific class information which improves understanding of the disease. In addition to providing disease explainability, the difference maps also reveal how the model makes its decision: the condition with the least reconstruction error is selected as the predicted class.

### 4.4. Uncertainty Quantification

The uncertainty quantification of diffusion classifiers is demonstrated in Figure 2(b). In addition to competitive classification performance and intrinsic explainability, uncertainty quantifications can be estimated without any model modifications. In medical imaging, uncertainty measures are validated by confirming that when the model is confident, the

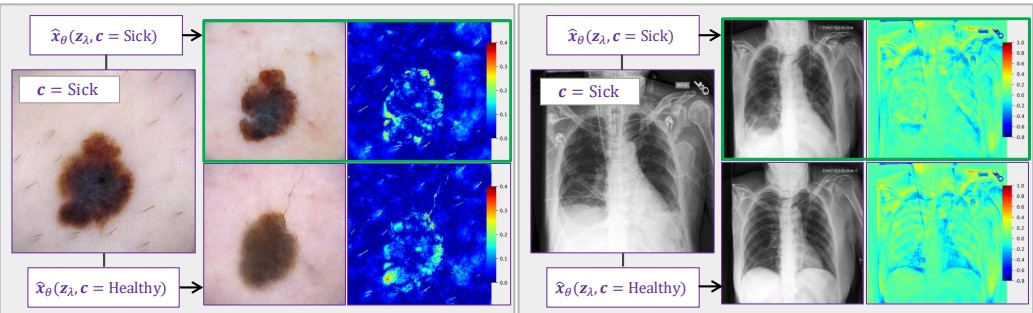

Figure 3: **Diffusion classifiers are naturally explainable** and highlight why they make classification decisions using classifier-free guided sampling. Difference maps show conditional areas of interest (pathology added/removed) during reconstruction.

prediction is correct, and when it is incorrect, it is uncertain (Nair et al., 2020). In terms of clinical decision making, quantifying this behaviour supports the idea that high-confidence predictions from the model are more trustworthy, while uncertain cases can be flagged for further testing or expert review. We therefore validate the diffusion model's uncertainty quantification by filtering out the most uncertain predictions and examining the change in performance on the remaining samples. The models show that accuracy increases when the most uncertain predictions are filtered out for CheXpert (- -) and ISIC (-). This confirms the effectiveness of their uncertainty measure and potential value across medical applications. We show that this same phenomenon holds with the trained DiT and pre-trained Stable Diffusion classifiers in Appendix D.

## 5. Conclusion

In this paper, we provide a comprehensive examination of the benefits of diffusion classifiers in medical imaging. First, we introduce a novel majority voting method to improve the overall performance of diffusion classifiers. We next demonstrate that diffusion classifiers are able to achieve comparable performance to state-of-the-art discriminative classifiers, in addition to providing intrinsic counterfactual explainability and uncertainty quantification.

Future work can extend our study to assess the robustness of diffusion classifiers in medical imaging, particularly under domain shifts or variations in image acquisition protocols. In addition, diffusion classifiers should be evaluated on more complex medical image classification scenarios, including 3D image classification and multi-class classification. Further, given that we present a novel uncertainty estimation, an in-depth analysis against other uncertainty methods using common metrics (i.e., reliability plots, failure analysis) should be performed.

Due to the nature of classifying images by accumulating a series of predictions, conditional diffusion models are limited by inference speed and computational requirements. To provide a reference, classifying a single batch of 128 images ($256^2$) takes 3:48 minutes with the UNet diffusion classifier on an A100 GPU. We provide a more thorough breakdown of computational requirements in Appendix F.

## Acknowledgments

We thank Bruno Travouillon, Olexa Bilaniuk, and the Mila IDT team for their support with Mila's HPC. This work was supported by the Natural Sciences and Engineering Research Council of Canada, Fonds de Recherche du Quebec: Nature et Technologies, the Canadian Institute for Advanced Research (CIFAR) Artificial Intelligence Chairs program, Google Research, Calcul Quebec, the Digital Research Alliance of Canada, the Vadasz Scholar McGill Engineering Doctoral Award, and Mila - Quebec AI Institute.

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

## Appendix A. Additional Background on Diffusion Classifiers

### A.1. Variational Diffusion Model Formulation

In the variational diffusion model formulation (Kingma et al., 2023), a noised image, $z_t$, can be generated from an uncorrupted image, $x$, at any point in a forward process marked by a continuous valued noise level, $t \sim [0, 1]$:

$$z_t = \alpha_\lambda x + \sigma_\lambda \epsilon \text{ where } \epsilon \sim \mathcal{N}(0, 1)$$

When noise is added to $x$, we are effectively decreasing its signal-to-noise ratio (and log-SNR, $\lambda$). There is a mapping that exists between the noise level, $t$, and the log-SNR, $\lambda$, called the noise schedule, $f_\lambda(t)$. For example, with a shifted-cosine noise schedule, this mapping is $\lambda = f_\lambda(t) = -2 \log \tan(\pi t/2) + 2 \log(\frac{64}{256})$. Therefore, when diffusing an image, we first sample $t \sim \mathcal{U}(0, 1)$ and then use this value and the noise schedule to calculate $\lambda$. There are many different noise schedules that differ in the way they modulate the log-SNR at various values along the range $[0, 1]$, though they must be strictly monotonically decreasing.

To corrupt the image, we ultimately need the values of $\alpha_\lambda$ and $\sigma_\lambda$, which we can derive with two additional pieces of information. First, we set the log-SNR as $\lambda = \log \alpha_\lambda^2 / \sigma_\lambda^2$. Additionally, we assume that the forward process itself is *variance preserving*, which implies $\alpha_\lambda^2 + \sigma_\lambda^2 = 1$. Armed with these two relations, we can derive $\alpha_\lambda^2$ as:

$$\lambda = \log \alpha_\lambda^2 / \sigma_\lambda^2$$
$$e^\lambda = \alpha_\lambda^2 / (1 - \alpha_\lambda^2)$$
$$\alpha_\lambda^2 = e^\lambda - e^\lambda \alpha_\lambda^2$$
$$\alpha_\lambda^2 (1 + e^\lambda) = e^\lambda$$
$$\alpha_\lambda^2 = \text{sigmoid}(\lambda)$$

And similarly, we can derive $\sigma_\lambda^2 = \text{sigmoid}(-\lambda)$.

### A.2. Derivation of Diffusion Classifier from Bayes' Rule

We provide a derivation of the diffusion classifier objective.

$$p_\theta(c_i | \boldsymbol{x}) = \frac{p_\theta(\boldsymbol{x}, c_i)}{\sum_{c_j} p_\theta(\boldsymbol{x}, c_j)}$$
$$= \frac{p_\theta(\boldsymbol{x} | c_i) p_\theta(c_i)}{\sum_{c_j} p_\theta(\boldsymbol{x} | c_j) p_\theta(c_j)}.$$

We assume the case in which we have no prior information about the relative frequencies of different classes and make the simplifying assumption that all classes are equally likely. Assuming a uniform prior over the labels, i.e., $p_\theta(c_i) = p_\theta(c_j)$ for all $i, j$, the prior cancels out in the fraction:

$$p_\theta(c_i | \boldsymbol{x}) = \frac{p_\theta(\boldsymbol{x} | c_i)}{\sum_{c_j} p_\theta(\boldsymbol{x} | c_j)}$$
$$= \frac{e^{\log p_\theta(\boldsymbol{x} | c_i)}}{\sum_{c_j} e^{\log p_\theta(\boldsymbol{x} | c_j)}}.$$

Using the variational diffusion model formulation, we approximate the likelihood as:

$$\log p_\theta(\boldsymbol{x}|c_i) \approx \mathbb{E}_{\epsilon \sim \mathcal{N}(0,\mathrm{I}), \lambda \sim p(\lambda)}[\|\boldsymbol{x} - \hat{\boldsymbol{x}}_\theta(\boldsymbol{z}_\lambda, c_i)\|_2^2].$$

Thus, the posterior probability simplifies to:

$$p_\theta(c_i|\boldsymbol{x}) = \frac{\exp\{\mathbb{E}_{\epsilon \sim \mathcal{N}(0,\mathrm{I}), \lambda \sim p(\lambda)}[\|\boldsymbol{x} - \hat{\boldsymbol{x}}_\theta(\boldsymbol{z}_\lambda, c_i)\|_2^2]\}}{\sum_{c_j} \exp\{\mathbb{E}_{\epsilon \sim \mathcal{N}(0,\mathrm{I}), \lambda \sim p(\lambda)}[\|\boldsymbol{x} - \hat{\boldsymbol{x}}_\theta(\boldsymbol{z}_\lambda, c_j)\|_2^2]\}}.$$

### A.3. Majority Voting Algorithm

We provide the pseudocode for the diffusion classification algorithm used in the experiments. We opt for a majority voting scheme as opposed to the average reconstruction error approach outlined by (Li et al., 2023).

```
def classify(x, num_classes, classification_steps):
    errors = fill((x.shape[0], num_classes, classification_steps), float('
        inf'))
    for step in classification_steps:
        t = rand(0, 1)
        z_t, eps_t = diffuse(x, t)  # add noise to image at t
        # Get the errors for each class
        for c in range(num_classes):
            pred = model(z_t, t, c)  # get noise prediction for given class
            error = mse(pred, eps_t)
            errors[:, c, step] = error  # store the error

    # Find the class with the lowest error for each step
    end_of_stage_votes = errors[:, :, :classification_steps].argmin(dim=1)

    # Count the votes for each class across all steps
    votes = zeros(x.shape[0], num_classes)
    for b in range(x.shape[0]):
        for step in range(classification_steps):
            class_with_lowest_error = end_of_stage_votes[b, step]
            votes[b, class_with_lowest_error] += 1

    final_classes = votes.argmax(dim=1)

    return final_classes
```

## Appendix B. Experimental Details

### B.1. Diffusion Classifier Optimization Settings

We hold our optimization settings constant across all diffusion models trained for our experiments. A detailed summary is found in Table 3.

| Setting | Diffusion Model ($3\times256\times256$) |
|---|---|
| Batch Size | 128 |
| Optimizer | Adam |
| Learning Rate | $1 \times 10^{-4}$ |
| Learning Rate Warmup Steps | 250 |
| Gradient Clipping | 1.0 |
| EMA Beta | 0.999 |
| EMA Warmup Steps | 50 |
| EMA Update Frequency | 5 |

Table 3: Optimization settings for our conditional diffusion models

### B.2. Discriminative Baseline Optimization Settings

We use official implementations of ResNet-based (`torchvision`), EfficientNet- and ViT-based (`timm`) classifiers in our experiments. A detailed summary of optimization settings for our discriminative baselines is found in Table 4.

| Setting | RN/EN (ISIC) | RN/EN (CheXpert) | ViT/Swin |
|---|---|---|---|
| Batch Size | 64 | 64 | 64 |
| Optimizer | AdamW | AdamW | Adam |
| Learning Rate | $1 \times 10^{-4}$ | $1 \times 10^{-4}$ | $1 \times 10^{-5}$ |
| Weight Decay | $1 \times 10^{-5}$ | $1 \times 10^{-3}$ | — |
| **Data Augmentation** | **Notes** | | |
| Random Rotation | Degree range: (-30, 30) | | |
| Random Horizontal Flip | Probability: 0.5 | | |
| Random Vertical Flip | Probability: 0.5 | | |
| Random Gaussian Blur | Kernel size: 5, Sigma range: (0.1, 2) | | |

Table 4: Optimization settings for discriminative baselines.

### B.3. UNet Settings

The ADM architecture (Dhariwal and Nichol, 2021) is used as a starting point, with minor alterations based on capacity requirements of each experiment. Class conditions are integrated into the model using cross-attention with a trainable module `nn.encoder`. A detailed summary is found in Table 5.

### B.4. DiT Settings

The DiT-B/4 architecture is followed as presented in (Peebles and Xie, 2023). A detailed summary is found in Table 6.

| Setting | UNet Model (3×256×256) |
| --- | --- |
| Prediction Parameterization | velocity |
| Noise Schedule | Shifted Cosine, Base-64 |
| Wavelet Transform | Single-stage Haar Wavelet |
| Sample Size | 128 |
| Channels | 12 |
| ResNet Layers per Block | 2 |
| Base Channels | 128 |
| Channel Multiplier | (1, 1, 2, 4, 8) |
| Cross Attention Resolution | 16 |
| Encoder Type | nn |
| Cross Attention Dimension | 512 |

Table 5: Settings for UNet model.

| Setting | DiT Model (3×256×256) |
| --- | --- |
| Prediction Parameterization | velocity |
| Noise Schedule | Shifted Cosine, Base-64 |
| Wavelet Transform | Single-stage Haar Wavelet |
| Sample Size | 128 |
| Channels | 12 |
| Number of Attention Heads | 12 |
| Attention Head Dimension | 64 |
| Number of Layers | 12 |
| Patch Size | 4 |

Table 6: Settings for DiT model.

## Appendix C. Stable Diffusion v2 Fine-Tuning

We fine-tune Stable Diffusion v2-base (Rombach et al., 2022) using the `Hugging Face` training pipeline for a total of 15k iterations. We construct the fine-tuning dataset by amalgamating our CheXpert and ISIC Melanoma training splits. Given that the model is designed for text-to-image generation, we replace labels in the datasets with text prompts, ie. "a benign skin lesion", or, "a frontal chest xray of a sick patient with pleural effusion". Fine-tuning dramatically increased Stable Diffusion's domain knowledge and subsequent classification performance on our benchmark datasets.

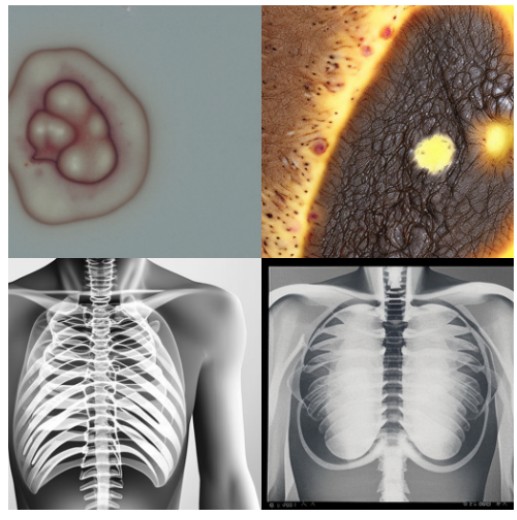
(*a*) No fine-tuning

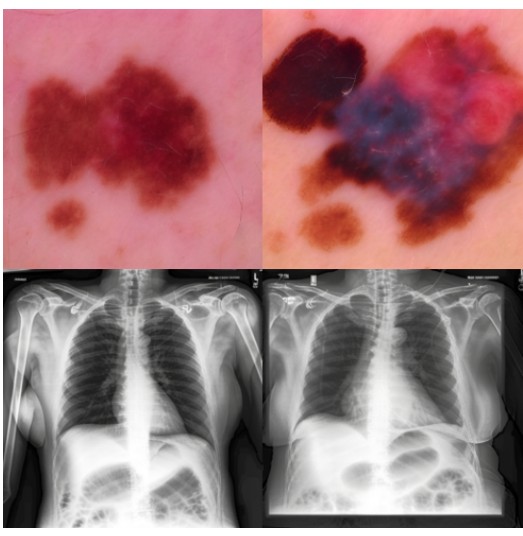
(*b*) After fine-tuning for 15k steps

Figure 4: Task-related generation from the Stable Diffusion v2-base model before (left) and after (right) fine-tuning. Training for only a few thousand iterations dramatically increased in-distribution inference and classification performance.

## Appendix D. Uncertainty Quantification

We validate our model uncertainty through measuring performance as uncertain predictions are filtered out. For both the pre-trained Stable Diffusion classifier and our diffusion classifiers trained from scratch, accuracy increases for both datasets as the most uncertain predictions are filtered out. This indicates that the model is most uncertain about its incorrect predictions, which is highly valuable across medical applications. See Figure 5 for a breakdown of this quantification in boxplot form.

## Appendix E. More Explainability Results

More explainability results can be found in Figure 6, and Figure 7. Input sick images have been altered to healthy class by adding noise to the input image and denoising with the healthy class. For CheXpert t=0.5 and for ISIC t=0.3 are used. CFG scale is 7.5.

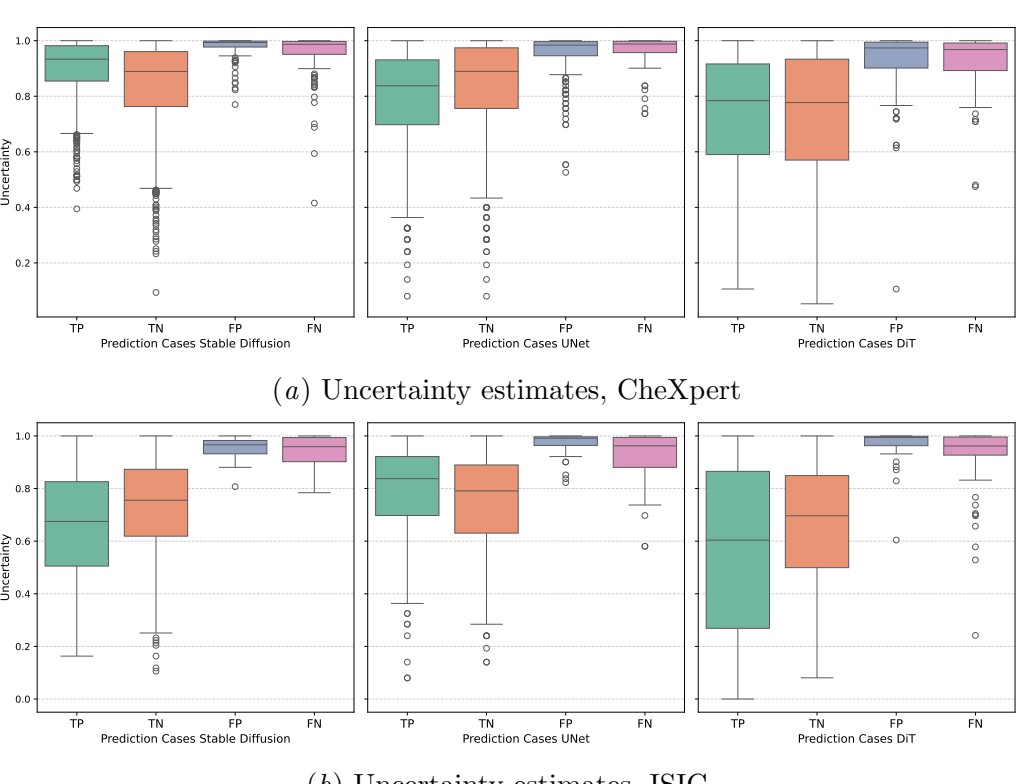

(*a*) Uncertainty estimates, CheXpert

(*b*) Uncertainty estimates, ISIC

Figure 5: We find that all of our diffusion classifier models are more confident about their correct predictions (TP, TN) than their incorrect predictions (FP, FN).

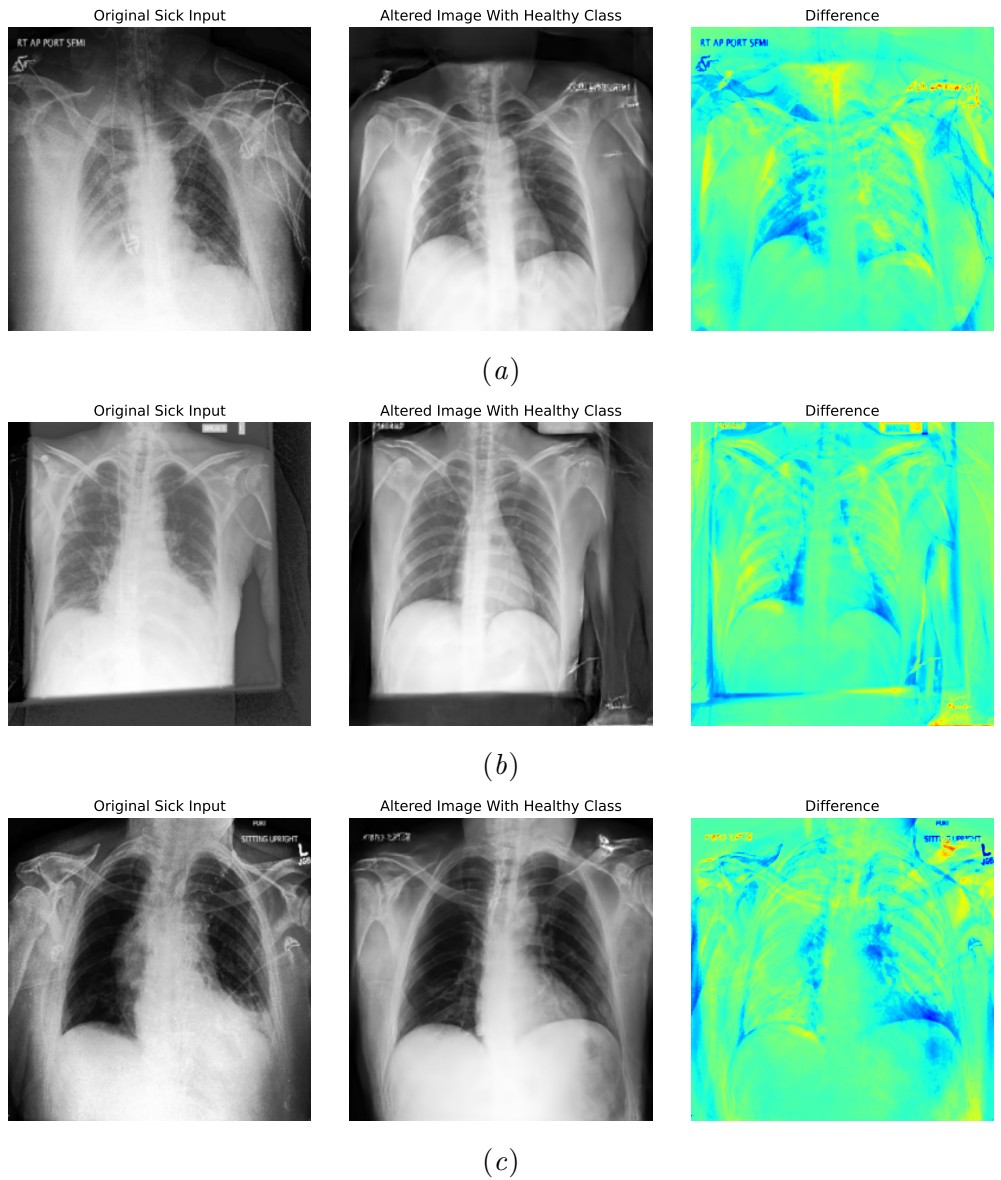

Figure 6: More explainability results for CheXpert by converting input sick images to healthy images. t=0.5 and CFG=7.5 are used for generating these images.

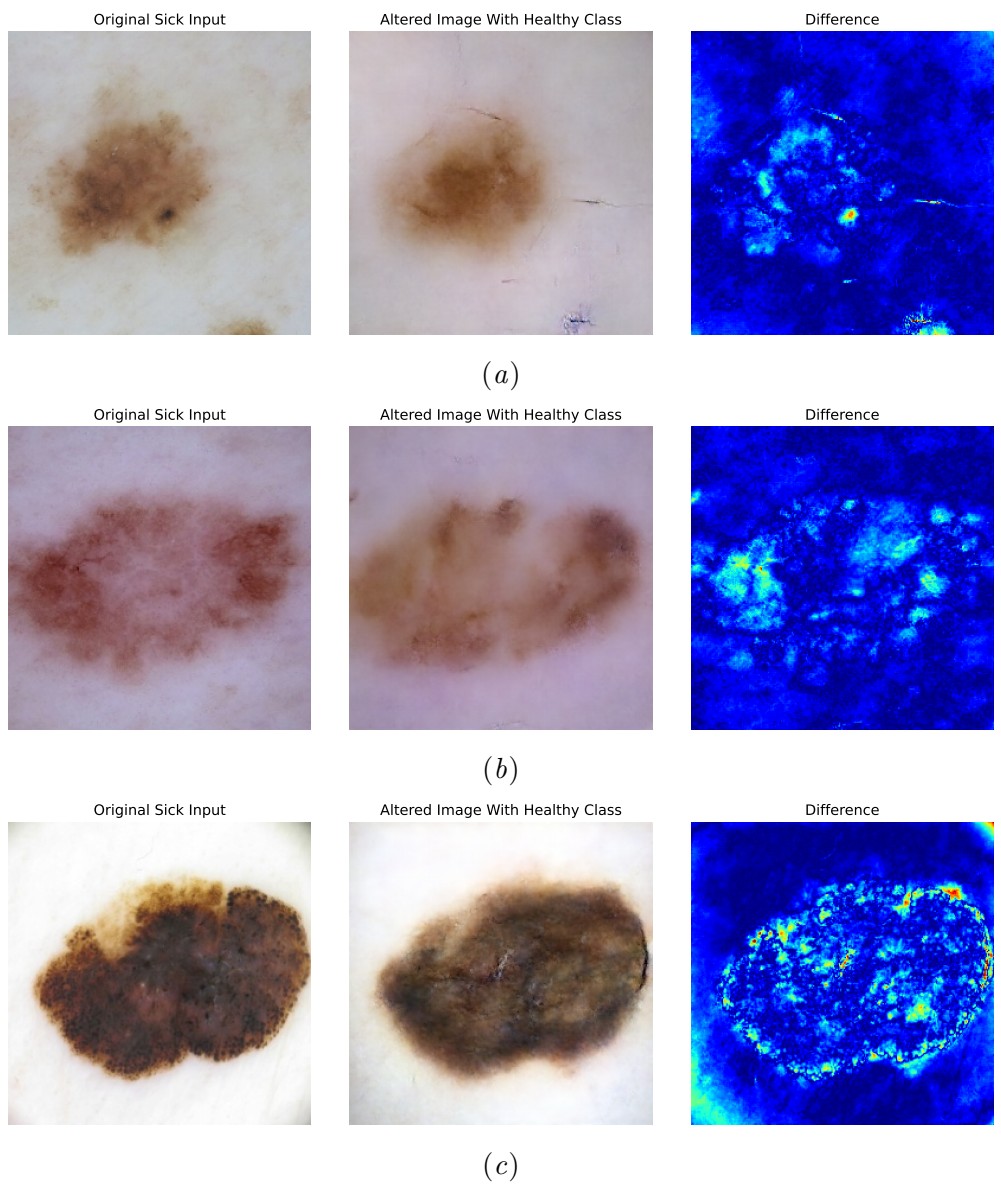

Figure 7: More explainability results for ISIC by converting input sick images to healthy images. t=0.3 and CFG=7.5 are used for generating these images.

## Appendix F. Computational Resources

All models were trained or fine-tuned a compute cluster of 80 GB A100 GPUs for all experiments in this paper. For inference, a single 80 GB A100 GPU is used. We provide a full breakdown of parameter count and time to classify a batch of 128 images in Table 7.

| Model | Parameters | Time per Batch (s) |
|---|---|---|
| UNet | 276M | 228 |
| DiT-B/4 | 148M | 195 |
| SD v2-base | 866M | 269 |
| ResNet-18 | 12M | 0.011 |
| ResNet-50 | 26M | 0.031 |
| EfficientNet-B0 | 5M | 0.020 |
| EfficientNet-B4 | 19M | 0.050 |
| ViT-B/16 | 87M | 0.036 |
| ViT-S/16 | 22M | 0.016 |
| Swin-B | 88M | 0.090 |

Table 7: Comparison of computational cost and inference speed across different models in our experiments. Parameter count is provided in millions (M) and inference time is provided in seconds (s) for a single batch of size 128 images.

