# OpenReview forum: "Conditional Diffusion Models are Medical Image Classifiers that Provide Explainability and Uncertainty for Free"
_MIDL.io/2025/Conference — MIDL 2025 Oral_

### Official Review · Reviewer_8BNj · 2025-02-11

**Confidence:** 4
**Preliminary Rating:** 2
**Recommendation:** Poster
**Final Rating:** 3

**Summary:**

The authors propose using a class-conditional diffusion model as a classifier instead. They inverse the generation process and reformulate the problem of generating images from noise as a classification problem by measuring the difference between ground truth and prediction. Their main contribution is using a voting scheme instead of an accumulation scheme, which results in improved performance. They evaluate their model on ISIC and Chexpert and demonstrate how the resulting model is inherently interpretable and can assess prediction confidence.

**Strengths:**

The paper's structure is very nice. It is nicely written, easy to follow, and clearly and understandably demonstrates many parts of its contribution.

The authors investigate a great variety of diffusion model architectures.

Experiments were conducted on multiple modalities at once.

The idea of using generative models for discriminative tasks is in line with current research, and therefore, the paper investigates an important line of research.

The paper's presentation is very good. Many experiments, like the confidence experiment in Figure 4., are immediately convincing.

**Weaknesses:**

1. Wrong claims: In the abstract you mention that discrimnative approaches need careful design, augmentation, and other training techniques and you claim that diffusion models do not need that. I believe that this is not true. Your tasks, which is a binary classification task on a balanced dataset, is also very restrictive and does not outperform discriminative approaches.

2. The Diffusion Model-based approaches for conditional image generation are not up to date. I think it would be interesting to investigate how well your approach works on current LDM approaches like EDM-2 or EDM-2-AG [1,2]. Approaches like ADM in image space are not up to date.

3. I don't think that a difference of 5.5 percentage points can be considered comparable. Right now I believe ResNet18 is the best model which questions the difficulty of the task in general and how well this approach generalizes to multi-class problems.

4. Lack of comparison to previous work. I would have loved to see some experiments on how your approach compares to other diffusion classifiers as well such as Chen et al. 2024.


[1]: Karras: Analyzing and Improving the Training Dynamics of Diffusion Models
[2]: Karras: Guiding a Diffusion Model with a Bad Version of Itself

**Detailed Comments:**

The majority vote should be described in more detail in the main paper and not hidden in the supplementary material.

Details, like the number of diffusion steps during training (I assume T = 1000), are missing from Figure 2.

Can you explain why you perform DWT on the images to train the diffusion model?

**Justification Of The Final Rating:**

While I still believe that the paper lacks detail and comparison to previous publications, the authors have nicely addressed some of my concerns in the rebuttal, which is why I am increasing my final rating.

**Justification Of The Preliminary Rating:**

At its current status, the proposed method's disadvantages outweigh its benefits. While the experiments demonstrate explainability and knowledge about uncertainty, the proposed method does not reach baseline performance and lacks comparison to previous work.

**Questions To Address In The Rebuttal:**

I am not entirely what the advantage of your method is compared to a classifier. High training and inference time paired with slightly worse performance currently outweigh the model's advantage in being naturally explainable and able to produce uncertainty.

**Special Issue:**

No

---

> ### Author Response · Authors · 2025-03-08
>
> We thank the reviewer for their thoughtful feedback.
>
> **The advantage of the method compared to a classifier:** We thank the reviewer for the opportunity to clarify our method’s strengths. This work is the first to explore diffusion classifiers for medical imaging. We do not claim that diffusion classifiers always outperform the best baselines which are trained explicitly for the task. Rather, we show that conditional diffusion models can be repurposed for classification in medical imaging and be competitive at worst and, in some cases, outperform popular classifiers that are trained for the particular task. Importantly, we show that conditional diffusion models offer several major advantages suited for medical imaging (explainability, uncertainty for free) that are not achievable with classifiers alone.
>
> **Description of majority voting algorithm in the main paper:** We appreciate the reviewer’s feedback on our majority voting pseudocode in Appendix A.2. In Sect. 2.3, we provide a detailed description of our majority voting algorithm and direct readers to Appendix A.2 for a more comprehensive explanation.
>
> **Range of diffusion training timesteps:** The reviewer points out that we do not state a discrete number of diffusion steps that our models were trained on. We clarify that we train our models using the variational diffusion model approach, in which $t$ is sampled from a continuous, uniform distribution between 0 and 1 (rather than a discrete number of timesteps). We improve the clarity of this in Sect. 2.1 of our manuscript.
>
> **Why is the DWT used?:** We thank the reviewer for allowing us to explain why we train our diffusion models in the wavelet domain rather than in pixel-space. By leveraging the discrete wavelet transform, we are able to halve the height and width of input images, losslessly, as a natural byproduct of the transform. In doing so, we compress images into a more computationally efficient form; we highlight this choice in Sect. 3.3.
>
> **Reviewer disputes claim: Discriminative approaches need careful design, augmentation, and other training techniques but not diffusion models:** It is established in the literature [1] that classifier performance can be highly sensitive to hyperparameters and data augmentation during training. As we propose a general framework for converting any conditional diffusion model into a classifier, the benefits of a simple and stable MSE loss function and strong regularization during training are inherited. Empirically, we find this to be true; conditional diffusion models were much less sensitive to hyperparameter selection during training than the baselines and achieve all results without data augmentations. Our optimization settings in Tables 4 and 5 of our updated manuscript.
>
> **Diffusion model approach:** The reviewer questions why EDM-2, a SOTA diffusion model in natural imaging, was not used in our experiments. Our method is a general framework for converting any conditional diffusion model (latent- or pixel-space) into a classifier that has explainability and uncertainty for free. Given the abundance of diffusion approaches, it would be challenging to compare all of them. In our trained from scratch models, we adopt the simple diffusion formulation and incorporate class conditioning using cross-attention or adaptive layer normalization. We additionally compare two widely used backbones—UNet and DiT, and a foundation model (Stable Diffusion).
>
> **ResNet18 is the best model:** We appreciate the reviewer’s thoughtful analysis of model performance and task complexity. While the diffusion classifiers have a 5.5% difference in performance in CheXpert compared to ResNet18, they perform as well as SOTA classifiers (ViT), and outperform ResNet18 on ISIC. We strive to show competitive performance of diffusion classifiers while incorporating additional features that are relevant to clinical tasks (e.g., explainability, uncertainty). We select a binary task to demonstrate this point and agree that further analysis on more complex tasks would be a useful future analysis (Sect. 5).
>
> **Lack of comparison to previous diffusion classifiers:** We reinforce that our work is the first of its kind in medical imaging. Other diffusion classifiers developed for natural images, such as [2], demonstrate robustness to various added perturbations. We have extended Sect. 5 to highlight this avenue in the future. Other prior work [3] focus on zero-shot classification using Stable Diffusion, which we compare in our paper.
>
> [1] He, K., et al. “Deep Residual Learning for Image Recognition.” 2016 IEEE Conference on Computer Vision and Pattern Recognition (CVPR), IEEE.
>
> [2] Chen, H., et al. "Diffusion models are certifiably robust classifiers." Advances in Neural Information Processing Systems 37 (2024): 50062-50097.
>
> [3] Krojer, B., et al. "Are diffusion models vision-and-language reasoners?." Thirty-seventh Conference on Neural Information Processing Systems. 2023.

---

> ### Comment · Area_Chair_ZRoL · 2025-03-13
> **discussion period ends in 2 days**
>
> A kind reminder that you now have 2 days to engage in discussion with the authors, with the aim of asking them any question about things that may remain unclear after their rebuttal.
>
> Please don't forget to fill in your final rating, even if it's the same rating.
>
> Thanks,
>
> MIDL 25

---

> > ### Comment · Area_Chair_ZRoL · 2025-03-14
> >
> > Hi, you now have around 24 hours to finalise your rating in the "final rating" section. Please do so even if you decide to keep your initial score. You may optionally respond to the authors rebuttal.
> >
> > Thanks
> >
> > MIDL 25

---

### Official Review · Reviewer_mK3A · 2025-02-25

**Confidence:** 5
**Preliminary Rating:** 4
**Recommendation:** Oral
**Final Rating:** 5

**Summary:**

This paper implements the idea of "diffusion model as a zero-shot classifier" in the medical domain and further illustrates its inherent explainability and uncertainty measurement. It introduces MC sampling for the noise level as a pair  ($\epsilon,\lambda$)and tests the model on two datasets: ISIC and CheXpert. The results seem promising.

**Strengths:**

1. this paper is well-written, clear, and easy to follow. As a reader, I had a smooth experience understanding the main idea.
2. the experiments conducted appear solid and comprehensive.
3. This work presents an interesting and meaningful attempt to integrate explainability and uncertainty measurement into a classifier-like (but diffusion) model.

**Weaknesses:**

I do not identify any major weaknesses in this paper.
However, there are some minor weaknesses that I hope the authors will address during the revision:

1. more literatures need to be included in the related work (which is integrated into the Introduction), particularly on **diffusion-based counterfactual explanation methods in the medical domain**. This should include, but is not limited to:
- Sanchez, P., Kascenas, A., Liu, X., O’Neil, A. Q., & Tsaftaris, S. A. (2022, September). What is healthy? generative counterfactual diffusion for lesion localization. In MICCAI Workshop on Deep Generative Models (pp. 34-44). Cham: Springer Nature Switzerland.
- Bedel, H. A., & Çukur, T. (2024). Dreamr: Diffusion-driven counterfactual explanation for functional mri. IEEE Transactions on Medical Imaging.
- Weng, N., Pegios, P., Petersen, E., Feragen, A., & Bigdeli, S. (2024, September). Fast diffusion-based counterfactuals for shortcut removal and generation. In European Conference on Computer Vision (pp. 338-357). Cham: Springer Nature Switzerland.
- Pegios, P., Lin, M., Weng, N., Svendsen, M. B. S., Bashir, Z., Bigdeli, S., ... & Feragen, A. (2024). Diffusion-based iterative counterfactual explanations for fetal ultrasound image quality assessment. arXiv preprint arXiv:2403.08700.

and, **diffusion-based uncertainty measurements in medical domain**(primarily in segmentation tasks) , including but not limited to:
- Amit, T., Shichrur, S., Shaharabany, T., & Wolf, L. (2023, October). Annotator consensus prediction for medical image segmentation with diffusion models. In International Conference on Medical Image Computing and Computer-Assisted Intervention (pp. 544-554). Cham: Springer Nature Switzerland.

2. how the pair of $(\epsilon, \lambda)$ be selected? I assume that $\epsilon$ is just randomly sampled from the gaussian distribution, is that correct? and what about $\lambda$? Under eq.4, "a identical set of .." how you select this identical set and it was selected from what?

3. In section 3.1, could you specify what is meant by "balancing the dataset"? Does this refer to class balancing, sample weighting, or some other form of balancing?

4. The result of stable diffusion seems not align across figure 2 (accu is approx. to 0.82) and table 1 (accu=0.85)

5.  It would be preferable to present hyperparameters in a table in the appendix for better readability and accessibility.

**Detailed Comments:**

Please see above under weakness

**Justification Of The Final Rating:**

The authors addressed all my concerns during the rebuttal and discussion phase. Additionally, I believe this work makes a strong contribution by implementing the idea of diffusion as a zero-shot classifier in the medical domain, effectively demonstrating the framework’s inherent explainability and uncertainty estimation capabilities. Therefore, I have updated my review to a strong accept.

**Justification Of The Preliminary Rating:**

This is a well-executed and interesting study, and I believe it is a highly relevant and promising contribution for MIDL. I provide some additional comments on minor weaknesses. Once these are addressed, I consider the paper to be ready for publication.

**Questions To Address In The Rebuttal:**

please refer to weakness

**Special Issue:**

Yes

---

> ### Author Response · Authors · 2025-03-08
>
> We thank the reviewer for their helpful and detailed feedback, and for their positive feedback on the strengths of the paper. The reviewer did not identify any major weaknesses with the paper but made some suggestions regarding additional references to add to the paper, in addition to  some detailed suggested improvements to the paper. We address these in turn.
>
> **Additional references** The reviewer pointed out relevant background work that adds to the body of literature existing for diffusion models in medical imaging. We updated the introduction of our manuscript to include these works, highlighting their applications in counterfactual generation and segmentation with uncertainty.
>
> **How the $(\epsilon, \lambda)$ pairs are selected**  We appreciate the opportunity to clarify this point: For each $(\epsilon, \lambda)$ pair, $\epsilon$ is sampled from an isotropic Gaussian distribution and $\lambda$ is sampled from $p(\lambda)$ (practically speaking, $t \sim \mathcal{U}(0,1)$, then $\lambda = f_\lambda(t)$). By “identical set”, we mean that each condition ($c_j$) is tested on the same $(\epsilon, \lambda)$ pair (sampled as described above) for a given image. We have added extra clarification to Sect. 2.2.
>
> **Section 3.1: how we balance the dataset** Indeed, this refers to class balancing. We have updated Sect. 3.1 to indicate as such.
>
> **The discrepancy between the accuracy values in Figure 2 (~0.82) and Table 1 (0.85)**
> We thank the reviewer for pointing out this discrepancy and we apologize for the confusion. In the revised manuscript, we have added a new table (Table 1, Sect. 4.1) comparing majority voting algorithm and average reconstruction error on the test set of both datasets, for each diffusion backbone (i.e., UNet, DiT, Stable Diffusion). Table 2 (Sect. 4.2 - previously Table 1) reports the performance of baseline classifiers and diffusion classifiers on the test set for both datasets. Figure 2 (Sect. 4.2) has been modified to show the step ablation on the validation split, which explains the accuracy differences between the figures and tables. The ablation study is conducted on the validation set to ensure that our model is not fine-tuned on the test set. While slight variations in accuracy are expected due to dataset composition differences, they do not affect our overall conclusions. This clarification has been added to the Figure 2 caption in the updated manuscript.
>
> **Present model hyper parameters and optimization settings in a table in the appendix** In our revised manuscript, we have formatted all model/optimizer settings in tables for better readability and accessibility. These tables are located in Appendix B in the updated manuscript.

---

> > ### Comment · Reviewer_mK3A · 2025-03-09
> >
> > Thanks for the detailed reply! I have no further questions regarding the references, dataset balance, or appendix—your response has already addressed my concerns.
> > However, I still have questions about how the $(\epsilon, \lambda)$pairs are selected. I reviewed the edited manuscript but couldn't find the definition of $ f_{\lambda}(t) $. Additionally, there are two definitions for $\lambda$ (in the paragraph after Eq. 1).
> > For the first definition,  $\lambda = \log \frac{\alpha_\lambda^2}{\sigma_{\lambda}^2}$
> > it seems that $\alpha_\lambda$ and $\sigma_{\lambda}$ are defined in terms of $\lambda$, which appears to be circular. Could you clarify:
> > - How is $ f_{\lambda}(t) $ defined?
> > - How should $\lambda$ be interpreted in light of this potential circular definition?

---

> > > ### Author Response · Authors · 2025-03-12
> > >
> > > Thank you for pointing this out! To clarify, we have updated the main text in Section 2.1: Diffusion Models (as shown below).
> > >
> > > In addition, we have provided a detailed derivation of $\alpha_\lambda$ and $\sigma_\lambda$ in Appendix A.1, which we display here as well. We hope that this eliminates the lack of clarity surrounding the definition of these variables.
> > >
> > > **Section 2.1: Diffusion Models**
> > >
> > > Following the variational diffusion model formulation (Kingma et al., 2023), the forward process is defined to be variance-preserving, imposing the constraint $\alpha_\lambda^2 = \text{sigmoid}(\lambda)$, $\sigma_\lambda^2=\text{sigmoid}(-\lambda)$, where $\lambda$ is the log-SNR given by $\lambda= f_\lambda(t)$ and $f_\lambda(t)$ is the noise schedule (see Appendix A.1). The noise schedule is a monotonically decreasing and invertible function that connects the time variable, $t$, with the log-SNR, $\lambda$. During training, $t$ is sampled from a continuous, uniform distribution, $\mathcal{U}(0,1)$, which is then used to compute $\lambda$. The resulting distribution over noise levels can be defined as $p(\lambda)=-1/f'_\lambda(t)$ (Kingma and Gao, 2023).
> > >
> > > **Appendix A.1**
> > >
> > > In the variational diffusion model formulation (Kingma et al., 2023), a noised image, $z_t$, can be generated from an uncorrupted image, $x$, at any point in a forward process marked by a continuous valued noise level, $t \sim [0,1]$:
> > >
> > > $$
> > >     z_t = \alpha_\lambda x + \sigma_\lambda \epsilon \text{ where } \epsilon \sim \mathcal{N}(0,1)
> > > $$
> > >
> > > When noise is added to $x$, we are effectively decreasing its signal-to-noise ratio (and log-SNR, $\lambda$). There is a mapping that exists between the noise level, $t$, and the log-SNR, $\lambda$, called the noise schedule, $f_\lambda(t)$. For example, with a shifted-cosine noise schedule, this mapping is $\lambda = f_\lambda(t) = -2 \log \tan (\pi t / 2) + 2 \log(\frac{64}{256})$. Therefore, when diffusing an image, we first sample $t \sim \mathcal{U}(0,1)$ and then use this value and the noise schedule to calculate $\lambda$. There are many different noise schedules that differ in the way they modulate the log-SNR at various values along the range $[0,1]$, though they must be strictly monotonically decreasing.
> > >
> > > To corrupt the image, we ultimately need the values of $\alpha_\lambda$ and $\sigma_\lambda$, which we can derive with two additional pieces of information. First, we set the log-SNR as $\lambda=\log \alpha_\lambda^2/\sigma_\lambda^2$. Additionally, we assume that the forward process itself is *variance preserving*, which implies $\alpha_\lambda^2+\sigma_\lambda^2=1$. Armed with these two relations, we can derive $\alpha_\lambda^2$ as:
> > >
> > > $$
> > > \lambda = \log \frac{\alpha_\lambda^2}{\sigma_\lambda^2}
> > > $$
> > > $$
> > > e^\lambda = \frac{\alpha_\lambda^2}{1 - \alpha_\lambda^2}
> > > $$
> > > $$
> > >  e^\lambda - e^\lambda \alpha_\lambda^2 = \alpha_\lambda^2
> > > $$
> > > $$
> > > \alpha_\lambda^2(1+e^\lambda) = e^\lambda
> > > $$
> > > $$
> > > \alpha_\lambda^2 = \text{sigmoid}(\lambda)
> > > $$
> > >
> > > And similarly, we can derive $\sigma_\lambda^2=\text{sigmoid}(-\lambda)$.

---

> > > > ### Comment · Reviewer_mK3A · 2025-03-12
> > > >
> > > > Thank you for the clarification! I have no further questions regarding this.
> > > > Good job! and good luck!

---

### Official Review · Reviewer_Y9qN · 2025-02-27

**Confidence:** 4
**Preliminary Rating:** 3
**Recommendation:** Poster
**Final Rating:** 4

**Summary:**

This paper demonstrates that conditional diffusion models (DM) trained for 2D image generation can be repurposed as medical image classifiers with comparable performance to traditional classification approaches. A novel majority voting scheme is introduced that improves classification performance. It is shown that DM can provide explainability through counterfactual generation and uncertainty estimation without additional modifications.
The presented approach is experimentally evaluated on the CheXpert and ISIC melanoma datasets. It is shown that majority voting is generally preferable to averaging and that the provided uncertainty correlates well with the accuracy of the classification.

**Strengths:**

* The paper introduces a novel approach to medical image classification with built-in explainability and uncertainty quantification.
* The proposed majority voting technique improves performance over the average reconstruction error approach.
* The intrinsic explainability via counterfactual generation is a clear advantage in the medical domain.
* The uncertainty quantification capabilities are well-demonstrated.
* Comparison to state-of-the-art classification models is performed.
* The paper is well-written and easy to follow.

**Weaknesses:**

* The experiments are limited to binary classification and the paper does not address an extension to multi-class classification.
* No statistical significance testing is conducted for the results of the majority voting.
* Lack of comparison to other explainability methods.

**Detailed Comments:**

* The derivation of Eq. (3) could be explained more clearly. A uniform prior over classes is used here because the authors used balanced datasets?
* Table 1 would benefit from error bars/repeated experiments and significance testing (however, I acknowledge that achieving highest classification accuracy was not the goal of this paper).
* Fig. 2: I would like to see the same plot for ISIC.
* How do the results from Fig. 4 relate to clinical decision making?
* Consider using a third dataset with multi-class classification.
* An analysis of failure cases would be interesting, e.g., where uncertainty is low but the prediction is incorrect.
* Evaluation of the uncertainty would benefit from using a proper scoring rule or the expected calibration error (ECE). Reliability diagrams would also be interesting.
* In Sect. 4.1 it is stated that majority voting improves the performance "across the board", but Fig. 2 shows that using the mean error is beneficial at lower classification steps. (maybe just rephrase that highest accuracy is consistently achieved with majority voting).

nitpicks:

* p. 5 "plural" instead of "pleural"
* Fig. 2 caption repeats itself ("leads to better performance ... leads to better classification")

**Justification Of The Final Rating:**

I thank the authors for addressing all my concerns. I welcome the additional information regarding Eq. (3), the repeated experiments, as well as the general explanations. The paper is a good fit for the conference and I therefore increase my rating.

**Justification Of The Preliminary Rating:**

While the core technique used in this paper is not novel, it presents novel approaches to using diffusion models as classifiers and introduces it to the field of medical imaging. It addresses important considerations for the medical domain and performs a valid analysis on two well-established datasets.
However, the majority voting is a core contribution of the paper and therefore, results on both datasets should be presented and discussed. I also miss an evaluation/comparison of the explainability feature against established methods.
If the authors manage to address that in the rebuttal, I'm willing to increase my score.

**Questions To Address In The Rebuttal:**

* What are the considerations for extending the presented method to multi-class classification?
* How does the explainability compare to established post-hoc methods, such as Grad-CAM?
* How does majority voting perform on the ISIC dataset? I did not find any results on that in the paper.

---

> ### Author Response · Authors · 2025-03-08
>
> We thank the reviewer for their insightful comments on the paper, and for their positive feedback on the strengths of the work. We first address their questions and then their detailed suggested improvements to the paper and revise our manuscript accordingly.
>
> **Extension to multi-class classification?:** We agree with the reviewer that it would be useful to understand the performance of diffusion classifiers in multi-class classification. This work presents the first use of conditional diffusion models as classifiers in medical imaging and we perform extensive experiments in two contexts to validate the work. Future work should extend it to multi-class classification, which we have addressed in Sect. 5. In addressing the reviewer’s questions about what is involved in this extension, one can build on prior work in natural images which demonstrate how diffusion classifiers' are used in multi-class settings. The main difference from binary classification is that the number of Function Evaluations (NFEs) scales with the number of classes, increasing inference time.
>
> **Lack of comparison to other explainability methods:**  The reviewer has requested a comparison of our explainability results with “established post-hoc methods, such as Grad-CAM.” However, we would like to clarify our reasoning for omitting any such comparison, which we felt is neither relevant nor feasible: (1) Grad-CAM vs. Counterfactual Explainability: Post-hoc methods like Grad-CAM highlight regions of an image that influenced a model’s decision but does not explain what would need to change to alter the classification. In medical imaging, counterfactual explanations—rooted in causality—have gained prominence [1], as they answer: “How would the image change if the class label were different?” Achieving this with a standard CNN would require an additional post-hoc conditional generative model, which is both complex and challenging. In contrast, our model, a conditional generative model, inherently answers this question. (2) Lack of a Meaningful Comparison Metric: Our model inherently provides an explanation during inference by providing both factual and counterfactual reconstructions, whose difference maps provide an interpretable visualization of the features driving classification decisions. The effectiveness of our approach lies in its transparency and ability to generate direct visual evidence for decision-making. Comparing this result to the explainability results from a different post-hoc method applied to the model would neither be feasible nor informative.
>
> **How does majority voting perform on the ISIC dataset?:** We’ve restructured the voting comparison results into Table 1 to directly compare majority voting to the average reconstruction error objective. We run repeated inferences on the test set using different random seeds and report mean accuracy with error bars for both algorithms. Additionally, we have included the steps ablation plot for the ISIC dataset in Figure 2.
>
> | Method               | CheXpert Majority | CheXpert Average | ISIC Majority | ISIC Average |
> |----------------------|------------------|------------------|--------------|--------------|
> | DiT-B/4             | 86.8 ± 0.38       | 85.3 ± 0.40       | 90.5 ± 0.08  | 90.4 ± 0.29  |
> | UNet                | 84.6 ± 0.21       | 83.4 ± 0.14       | 91.2 ± 0.08  | 84.3 ± 0.75  |
> | Stable Diffusion  | 84.7 ± 0.34       | 79.9 ± 0.29       | 94.7 ± 0.14  | 94.5 ± 0.33  |
>
> **Uncertainty and its relationship to clinical decision making:** We show that where the model is more certain, it is more likely to be correct, and where it is uncertain, it is more likely to be incorrect. In terms of clinical decision making, quantifying this behaviour supports the idea that high-confidence predictions from the model are more trustworthy, while uncertain cases can be flagged for further testing or expert review. We amend Sect. 4.4 in our revised manuscript to better clarify this.
>
> **Evaluation of the uncertainty via expected calibration error (ECE) or reliability diagrams:** While we focus on uncertainty-based sample selection in our work, we acknowledge the importance of calibration and failure case analysis in many applications and will consider exploring these ideas in future work (Sect. 5).
>
> **Additional clarity for derivation of Eq. 3:** In an effort to improve clarity, we have added a more complete derivation of the diffusion classifier objective (Eq. (3)) in Appendix A.1. We assume the case in which we have no prior information about the relative frequencies of different classes and make the simplifying assumption that all classes are equally likely.
>
> [1] Sanchez, Pedro, and Sotirios A. Tsaftaris. "Diffusion causal models for counterfactual estimation." arXiv preprint arXiv:2202.10166 (2022).

---

> ### Comment · Area_Chair_ZRoL · 2025-03-13
> **discussion period ends in 2 days**
>
> A kind reminder that you now have 2 days to engage in discussion with the authors, with the aim of asking them any question about things that may remain unclear after their rebuttal.
>
> Please don't forget to fill in your final rating, even if it's the same rating.
>
> Thanks,
>
> MIDL 25

---

### Official Review · Reviewer_RqkT · 2025-03-01

**Confidence:** 4
**Preliminary Rating:** 4
**Recommendation:** Oral
**Final Rating:** 5

**Summary:**

The paper investigates the use of diffusion models, not as a image generation tool, but as an classification tool, with (claimed) integrated interpretation and confidence/uncertainty estimation.

The results are quite interesting, and even if somewhat preliminary, point to a very promising research area for the field. Right now the clarity of the manuscript could still be improved.

**Strengths:**

- The paper is interesting and thought provoking;
- it is a good fit for the conference and timely, addressing an important difficulty of the field;
- the authors share their code;
- the figures are well made and useful to help comprehension of the paper (notably Figure 1).

**Weaknesses:**

- The way the notation is introduced could be streamlined, especially for people less familiar with (Kingma and Gao, 2023)
- the paper structure is a bit clumsy at times, notably due to the creative figures placement (see my suggestion in the detailed comment)

**Detailed Comments:**

Misc:
- a lot of space ends-up being wasted due to the long captions in the figures. The PMLR template has a `\hangindent` buried somewhere in the captions definition, which effectively increase the margins (and reducing text "real-estate"). The authors could reclaim precious space by balancing better between the figures caption (shorter and straight to the point) and its discussion (to keep within the main body of text)
- use `\eqref` and not vanilla `\ref` when referencing equations.
- for the code listing in the appendix, consider adding syntax coloring ( https://www.overleaf.com/learn/latex/Code_listing#Code_styles_and_colours has a somewhat ready made example)
- it may be relevant to move Fig. 6 into the main body of the manuscript

**Justification Of The Final Rating:**

I am happy with the rebuttal and discussion provided by the authors, and decided to increase my score. This paper would, in my opinion, make a very nice oral presentation at MIDL, to showcase this novel line of research.

**Justification Of The Preliminary Rating:**

The findings are novel and interesting, and while preliminary in many respects, very interesting for MIDL and its community. This is why I tentatively suggest an oral on top of the weak accept, and as well as a special issue recommendation.

**Questions To Address In The Rebuttal:**

### Discussion on inference
I would like to see more discussion on the inference, notably its speed, and memory/computational requirements, compared to "standard" networks. Not necessarily that it is bad if it is slower/more memory hungry, but this is an important nuance to get a full picture. Right now only one inference time (for 501 steps) is presented, without full context.

### Potential for 3D scans?
Right now only 2D tasks are tested. What is the potential for 3D scans, if computationally feasible at all?

### The paper could be restructured
Not fully a rebuttal question, but I think the presentation of the paper can be quite improved, to make it easier to read and sift through

**Special Issue:**

Yes

---

> ### Comment · Reviewer_RqkT · 2025-03-01
> **Note that this was an emergency review**
>
> I did not mention it in the review itself to not influence other reviewers and authors.

---

> ### Author Response · Authors · 2025-03-08
>
> We thank the reviewer for their valuable comments. We are very pleased that they find the paper novel and interesting for MIDL and the community. We appreciate the reviewer’s suggestions for improving the clarity and structure of the paper in a few places, as well as their suggestions for adding some details regarding inference metrics. We attempt to address all of the reviewer’s comments as thoroughly as possible.
>
> **Discussion on Inference:** As requested, we include more discussion and context on the inference speed and computational requirements. We add Table 7 to Appendix F and relevant discussion to Sect. 5. Note that parameter count is provided in millions (M) and inference time is provided in seconds calculated using a single 80 GB A100 GPU with a batch size of 128.
>
> | Model            | Params | Time per Batch (s) |
> |-----------------|--------|--------------------|
> | UNet           | 276M   | 228                |
> | DiT-B/4        | 148M   | 195                |
> | SD v2-base     | 866M   | 269                |
> | ResNet-18      | 12M    | 0.011              |
> | ResNet-50      | 26M    | 0.031              |
> | EfficientNet-B0| 5M     | 0.020              |
> | EfficientNet-B4| 19M    | 0.050              |
> | ViT-B/16       | 87M    | 0.036              |
> | VIT-S/16       | 22M    | 0.016              |
> | Swin-B         | 88M    | 0.090              |
>
> **Potential for 3D Scans?:** We agree with the reviewer that the exploration of diffusion classifiers for 3D medical images would be an interesting extension to our analysis. Diffusion models, including the ones used for our classifier, are inherently adaptable to 3D data. The key challenge lies in managing the significantly higher computational burden and memory requirements of 3D convolutions and feature maps compared to 2D. Whether this is feasible, and at what capacity, is highly dependent on the compute resources that a researcher has access to. For the sake of time and ease of iteration, we chose to run our experiments on 2D datasets in this paper, but highlight this possible extension in Sect. 5.
>
> **Restructuring the formatting and clarity:** We thank the reviewer for the detailed comments and suggestions regarding formatting. As suggested, we (1) updated references to equations to use the \eqref LaTeX command; (2) updated our majority voting algorithm pseudocode block to be color formatted. (3) While we attempted to keep figure captions concise, we note that they are intentionally comprehensive to ensure those skimming the paper can still understand key contributions without requiring a deep dive into the main text.

---

> > ### Comment · Reviewer_RqkT · 2025-03-12
> >
> > I thank the authors for their rebuttal, new work and additional numbers.
> >
> > For tonight (limited brain bandwidth) I want to focus on a key point (as a discussion, no worries about potential negative impact on score):
> >
> > > The key challenge lies in managing the significantly higher computational burden and memory requirements of 3D convolutions and feature maps compared to 2D
> >
> > But isn't the challenge for 3D data that it is effectively impossible, with current hardware, to process the whole 3D scan as a whole? Henceforth the need to sub-patch the 3D scan. (Which is obviously not required for 2D data, at least as long as you are not dabbling in histopathology.) So that leads me to the question: would diffusion classifiers be possible/make sense running on sub-patch of scans?

---

> > > ### Author Response · Authors · 2025-03-13
> > >
> > > Thank you for highlighting this.
> > >
> > > Indeed, as noted, a critical challenge when applying diffusion models directly to 3D data lies in the extensive computational and memory demands. Recent advances, however, provide promising strategies for addressing this constraint.
> > >
> > >
> > > Specifically, diffusion-based models utilizing wavelet transforms [1] have demonstrated the ability to operate directly within the voxel space, effectively managing computational load while retaining high-resolution synthesis capability. Additionally, leveraging latent diffusion models (LDMs) for 3D data ([2], [3]) significantly reduces memory and computational demands by performing diffusion steps in a compact latent space representation, rather than directly in voxel space.
> > >
> > >
> > > These approaches suggest that diffusion classifiers could indeed be feasibly and effectively applied to 3D scans, depending on the choice of representation (wavelet domain or latent space). Such methods could effectively alleviate the computational limitations mentioned. As our methodology uses the wavelet domain to reduce computational load, we believe extending it to 3D data should be straightforward.
> > >
> > >
> > > **References**:
> > >
> > > [1] Friedrich, P., et al. (2024). "WDM: 3D Wavelet Diffusion Models for High-resolution Medical Image Synthesis." MICCAI Workshop on Deep Generative Models, Springer Nature.
> > >
> > > [2] Pinaya, W.H.L., et al. (2022). "Brain Imaging Generation with Latent Diffusion Models." MICCAI Workshop on Deep Generative Models, Springer Nature.
> > >
> > > [3] Zhu, L., et al. (2023). "Make-A-Volume: Leveraging Latent Diffusion Models for Cross-modality 3D Brain MRI Synthesis." International Conference on Medical Image Computing and Computer-Assisted Intervention, Springer Nature.

---

> > > > ### Comment · Reviewer_RqkT · 2025-03-15
> > > >
> > > > Very interesting, thank you for the detailed explanation. Curious to see what lies ahead in the coming years :-)

---

> ### Comment · Area_Chair_ZRoL · 2025-03-14
>
> Hi, this is kind reminder that you now have around 24 hours to finalise your rating in the "final rating" section. Please do so even if you decide to keep your initial score. You may optionally respond to the authors rebuttal.
>
> Thanks
>
> MIDL 25

---

> ### Comment · Area_Chair_ZRoL · 2025-03-15
> **a few hours left to update your rating**
>
> please don't forget to update your final score for this paper, you only have a few hours left :)

---

### Comment · Area_Chair_ZRoL · 2025-03-06
**Rebuttal deadline reminder**

A kind reminder that the deadline to submit your response is March 7 23:59 AoE, so you have around 44 hours as of now.

Good luck for the final push!

MIDL 25

---

### Author Rebuttal · Authors · 2025-03-08

**Rebuttal:**

Please find our revised manuscript (9 pages, changes in red text) within the .zip file identified as Supporting Material.

**Supporting Material:**

/attachment/d833225b35b1cad86e319beaa8beac81d528faf6.zip

---

### Author Response · Authors · 2025-03-08

We would like to thank all the reviewers for their valuable insights and thoughtful review of our paper. We appreciate the thoroughness with which each reviewer reviewed our manuscript, offering several insightful suggestions and highlighting a few areas of improvement.

We are pleased that the reviewers found our paper well-written, the area of research important and timely, and the experiments convincing overall. We will make our code public, for ease of implementation and reproducibility of our approach. We documented our code, making it easy to understand and to use.

We have responded to each reviewer’s comments individually below and would be happy to address any remaining comments or concerns. We addressed the main concerns and made the associated changes in the original manuscript (shown in red). This includes an additional experiment comparing averaging to majority voting (Table 1), the addition of the steps ablation plot for the ISIC Melanoma dataset (Figure 2) and other clarifications, as requested. Detailed changes are described below. We thank the reviewers for their suggestions, as we feel they helped improve the paper.

---

### Meta-Review · Area_Chair_ZRoL · 2025-03-20

**Recommendation:** Accept (Oral)
**Confidence:** 4

**Metareview:**

First, I would like to thank the authors and the reviewers for the constructive discussion.

All reviewers agree that this paper presents a novel idea, which is a promising line of research. Moreover, the claims are well supported by comprehensive experiments and ablation/sensitivity studies.

For these reasons I recommend acceptance, and I think the novel idea could be interesting for an oral presentation.

However, I would like the authors to make three changes to their manuscript:
1. add the justification provided to reviewer 4 about augmentation and hyper-parameter fine-tuning for discriminative vs. diffusion models.
2. I would like the authors to add statistical tests to ALL their experiments (with correction for multiple tests when necessary), to better support the advantage of the majority voting over averaging, as well as to get an idea of performance of the different models.
3. The authors have made substantial modifications to the original template. While this is not a strong violation of authors guidelines and doesn't prevent acceptance, I would like them to revert their manuscript to the provided template, possibly by moving some content to the appendix, or by applying some of reviewer 1's suggestions for space.